# Advancing Understanding of *Escherichia coli* Pathogenicity in Preterm Neonatal Sepsis

**DOI:** 10.3390/microorganisms13020219

**Published:** 2025-01-21

**Authors:** Oscar Villavicencio-Carrisoza, Orly Grobeisen-Duque, Ana Laura Garcia-Correa, Irma Eloisa Monroy-Muñoz, Graciela Villeda-Gabriel, Irma Elena Sosa-González, Hector Flores-Herrera, Ricardo Figueroa-Damian, Jorge Francisco Cerna-Cortes, Sandra Rivera-Gutierrez, Isabel Villegas-Mota, Veronica Zaga-Clavellina, Addy Cecilia Helguera-Repetto

**Affiliations:** 1Departamento de Inmunobioquímica, Instituto Nacional de Perinatología Isidro Espinosa de los Reyes, Ciudad de Mexico 11000, Mexico; cuauqbp@gmail.com (O.V.-C.); orly.grobeisen@gmail.com (O.G.-D.); floresh8@yahoo.com (H.F.-H.); v.zagaclavellina@gmail.com (V.Z.-C.); 2Departamento de Laboratorio Central, Instituto Nacional de Perinatología Isidro Espinosa de los Reyes, Ciudad de Mexico 11000, Mexico; lau.bio97@gmail.com; 3Departamento de Investigación en Salud Reproductiva y Perinatal, Instituto Nacional de Perinatología Isidro Espinosa de los Reyes, Ciudad de Mexico 11000, Mexico; irmae4901@gmail.com; 4Departamento de Infectología e Inmunología, Instituto Nacional de Perinatología Isidro Espinosa de los Reyes, Ciudad de Mexico 11000, Mexico; gracielavilleda@yahoo.com.mx (G.V.-G.); elenasosa9635@gmail.com (I.E.S.-G.); ricardo.figueroa@inper.gob.mx (R.F.-D.); 5Escuela Nacional de Ciencias Biologicas del Instituto Politecnico Nacional, Ciudad de Mexico 11350, Mexico; jorgecerna1008@gmail.com (J.F.C.-C.); san_rg@yahoo.com.mx (S.R.-G.); 6Secretaría de Salud del Estado de Quintana Roo, Chetumal 7700, Quintana Roo, Mexico; isabelvillegasmota@gmail.com

**Keywords:** neonatal sepsis, *E. coli* virulence, *E. coli* phylogroups, preterm birth sepsis, *E. coli* PFGE

## Abstract

Neonatal sepsis is a major cause of mortality in preterm infants, with *Escherichia coli* as one of the leading pathogens. Few studies have examined the interplay between virulence factors, resistance profiles, phylogroups, and clinical outcomes in this population. We analyzed 52 *E. coli* strains isolated from 49 preterm neonates diagnosed with sepsis at a tertiary-level hospital in Mexico. Strains underwent phylogenetic classification, virulence gene profiling, and antimicrobial resistance testing. PFGE was used to assess genetic relatedness and outbreak clusters. Clinical data were correlated with molecular findings. Phylogroups A and B2 accounted for 46% of strains. Phylogroup A exhibited notable virulence, with high prevalence of the pathogenicity island described in virulent extra-intestinal *E. coli* strains (PAI), aerobactin siderophore receptor AerJ (*iut*A), and yersiniabactin siderophore receptor (*fyu*A) genes, alongside significant resistance profiles. PFGE identified two dominating branches. Branch A, comprising phylogroups A and B2, displayed high resistance and was prevalent in the neonatal intensive care unit. Branch C, with phylogroups A and D, showed less multidrug resistance but was significantly associated with maternal chorioamnionitis. This study redefines *E. coli* pathogenicity in neonatal sepsis, highlighting the virulence of traditionally non-pathogenic phylogroups. High virulence strains were associated with more severe outcomes. These findings underscore the need for enhanced strategies in targeted prevention, improved diagnostics, and tailored treatments for high-risk preterm populations.

## 1. Introduction

*Escherichia coli* (*E. coli*) is a Gram-negative, facultative anaerobic bacterium that is commonly found as a part of the human gut microbiome. While generally harmless in its commensal form, *E. coli* is also a leading cause of extra-intestinal diseases such as urinary tract infection, soft tissue infection, newborn meningitis, and bacteremia. In their groundbreaking work, Clermont et al. developed a molecular classification system for *E. coli*, dividing it into four major phylogenetic groups: A, B1, B2, and D [1]. Group A and B1 are typically associated with commensal strains, while groups B2 and D are more commonly linked to pathogenic strains [2,3].

The extent of damage caused by an *E. coli* infection is influenced by both the host’s immune response and the bacterium’s genetic arraignment, specifically its virulence and resistance genes. These genes allow *E. coli* to survive, spread, and evade immune defenses contributing to its pathogenicity [2,4].

In recent years, the overuse and misuse of antibiotics have accelerated the emergence of multidrug-resistant bacteria, including *E. coli*. This resistance trend poses significant challenges, especially against commonly used antibiotic classes such as β-lactams and macrolides, thus pressuring healthcare systems to rely on more potent and costly alternatives [3,5,6]. By 2050, multidrug-resistant bacteria are projected to become a leading cause of mortality worldwide, with estimates suggesting up to 10 million lives could be claimed annually due to microbial resistance [7,8]. In response, the World Health Organization (WHO) has issued a global warning, ranking third-generation cephalosporin-resistant *E. coli* as the second most critical threat to be addressed in its strategic initiatives for infection prevention and control [9]. This rise in resistant *E. coli* strains, particularly in hospital settings, has led to an increase in severe infections such as bacteremia and sepsis, further jeopardizing the outcomes of vulnerable patients.

Sepsis, a systemic inflammatory response to infection, is often triggered by the spread of bacteria through the bloodstream. It is a life-threatening condition, particularly in vulnerable populations such as neonates, the elderly, pregnant people, and immunocompromised individuals [10,11]. Neonatal sepsis, in particular, remains a significant global health concern due to its high morbidity and mortality rates. Worldwide, 4 out of every 1000 live births result in neonatal sepsis, with mortality rates ranging from 25 to 33% of all neonatal deaths [12,13,14]. In Mexico, this incidence is even higher with rates between 4 and 15.4 per 1000 live births, surpassing those of other Latin American countries (1.1 to 8.6 per 1000 live births) [15,16]. These numbers highlight critical gaps in the prevention, diagnosis, and treatment of neonatal infections, particularly in resource-limited settings.

Risk factors for neonatal sepsis can be categorized into maternal or neonatal contributors. Common neonatal risk factors include prematurity, low birth weight, and the use of invasive medical devices such as catheters and endotracheal tubes. Maternal factors include chorioamnionitis, intra-amniotic infection, and vaginal colonization with group B streptococci (GBS) [12].

There are several agents that may cause neonatal sepsis. The most common pathogens seen in early-onset sepsis (EOS), typically transmitted vertically from the mother to the newborn, are 70% GBS and *E. coli*, with term newborns primarily affected by GBS (40–45%), while preterm newborns are more likely to contract *E. coli* (50%). Late-onset sepsis (LOS), which is usually acquired horizontally, is typically caused by Gram-positive bacteria like coagulase-negative staphylococci CNS (50%), *Staphylococcus aureus* (7%), and GBS (1%), while Gram-negative bacteria such as *E. coli*, *Klebsiella pneumoniae*, *Serratia marcescens*, *Enterobacter* spp., and *Pseudomonas aeruginosa* account for 40–52% of cases, with *E. coli* being the most common pathogen [13,14].

The most important factors making preterm neonates prone to neonatal sepsis caused by *E. coli* are their diminished innate and adaptative immune responses, coupled with the horizontal transmission of pathogens. In LOS, this transmission often occurs through invasive procedures such as the use of catheters or sample collection for cerebrospinal fluid, blood, or urine analysis. Despite *E. coli* being an intestinal bacterium primarily associated with extra-intestinal neonatal infections, its dissemination in hospital settings is facilitated through fomites, healthcare providers, and contact with other patients [17].

Healthcare-associated infections (HCAIs) are particularly prevalent in low- and middle-income countries (LMICs), disproportionately affecting vulnerable populations such as preterm and low-birth-weight neonates. According to the WHO in its “Global report on the epidemiology and burden of sepsis”, HCAIs are the leading cause of LOS. Among neonates who die in a hospital setting, 4–56% succumb to HCAIs, with the highest incidence rates reported in South East Asia and sub-Saharan Africa. Neonatal intensive care units (NICUs) in these regions report infection rates of 15.2 to 62 per 1000 patient days [18].

The dual challenges of multidrug-resistant *E. coli* strains and the immunological immaturity of preterm neonates create an environment ripe for infection. Neonates exhibit an underdeveloped immune system, characterized by Th2 cells’ polarized activity and underperforming Th1 cells’ response, alongside deficient up- and down-regulation of pro- and anti-inflammatory cytokines; these immune deficits render neonates highly susceptible to bacteremia and subsequent sepsis [17,19].

Given the high morbidity and mortality rates associated with neonatal sepsis due to *E. coli*, particularly in preterm neonates, it is critical to deepen our understanding of the factors contributing to its occurrence. In this study, we aim to analyze the patterns of neonatal sepsis in preterm neonates delivered in the Instituto Nacional de Perinatología Isidro Espinosa de los Reyes (INPer). Additionally, we will assess the phenotypic characteristics, as well as the virulence and resistance profiles of the bacterial strains responsible for these infections. This analysis is intended to provide valuable data that can be leveraged to inform the development of more effective prevention programs, enhance early diagnostic strategies, and optimize treatment protocols, ultimately contributing to a reduction in neonatal sepsis and improving outcomes in this vulnerable population.

## 2. Materials and Methods

### 2.1. Study Approval and Data Collection

This study was approved by the Research Ethics Committee of the Instituto Nacional de Perinatología (registration numbers 212250-3210-11007-04-14 and 212250-3210-20607-03-14). It is a retrospective, cross-sectional, observational study conducted at the Instituto Nacional de Perinatología between January 2017 and December 2019. Inclusion criteria included preterm neonates diagnosed with sepsis and confirmed *E. coli* isolation. Exclusion criteria included term neonates, neonates with clinical sepsis lacking microorganism identification, and those infected with a microbiological agent other than *E. coli*. Elimination criteria comprised cases initially diagnosed with *E. coli* but later identified as different organisms through molecular techniques.

Patient data were collected from electronic clinical files (the “Expediente Clínico Electrónico”, a legally required document in Mexico that includes comprehensive medical history). All patient information was anonymized prior to analysis.

Maternal variables considered in this study included cervicovaginitis, urinary tract infection, premature rupture of membranes, chorioamnionitis, and fever. Neonatal variables included hypothermia, tachycardia, bradycardia, tachypnea, bradypnea, meningitis, mortality, apnea, and birth weight. Birth weight was categorized based on the WHO classification as follows: normal birth weight (2500–3999 g), macrosomia (≥4000 g), low birth weight (LBW; <2500 g), very low birth weight (VLBW; <1500 g), and extremely low birth weight (ELBW; <1000 g) [20].

### 2.2. Biological Materials

*E. coli* strains were isolated from blood, urine, and cerebrospinal fluid (CSF), resulting in a total of 52 isolates identified. Bacterial identification and antimicrobial susceptibility testing were performed in the clinical microbiology laboratory using the Vitek 2^®^ system [BioMerieux SA, 69280 Marcy L’Etoile, Lyon, France], in accordance with Clinical and Laboratory Standards Institute guidelines CLSI-M100-ed33 [21] [Clinical and Laboratory Standards Institute, 19312 Berwyn, PA, USA]. *E. coli* cultures were grown in LB [Luria–Bertani; Beckton Dickinson and Company, 07417 Franklin Lakes, NJ, USA] medium and preserved in 15% glycerol [Merk KGaA, 64293 Darmstadt, Germany] at −70 °C. Genomic DNA was extracted following the manufacturer’s protocol using the Quick-gDNA™ MiniPrep kit [Zymo Research CORP, 92614 Irvine, CA, USA].

### 2.3. Multiresistance Classification

Multidrug resistance classification was based on the criteria established by WHONET 2023 and Clinical and Laboratory Standards Institute (CLSI) guidelines. The antibiotics analyzed included amikacin, amoxicillin/clavulanic acid, ampicillin, ampicillin/sulbactam, aztreonam, cefazolin, cefoxitin, cefepime, cefixime, ceftazidime, ceftriaxone, cefuroxime, cefuroxime axetil, ciprofloxacin, chloramphenicol, colistin, doripenem, ertapenem, gentamicin, imipenem, levofloxacin, meropenem, minocycline, moxifloxacin, nitrofurantoin, piperacillin, tetracycline, tigecycline, tobramycin, trimethoprim, and trimethoprim/sulfamethoxazole [BioMérieux SA].

Automated diagnostic panels [BioMérieux SA] were used for both blood and urine cultures, with automated antibiotic susceptibility testing specifically targeting Gram-negative bacteria. Resistance classification was performed following the guidelines using the WHONET platform [https://whonet.org accessed on December 2023]

### 2.4. Phylogroup Classification

*E. coli* isolates were classified into phylogenetic groups using 100 ng of DNA in the multiplex-PCR containing 0.125 μM of each primer and 0.6 U of Taq DNA Polymerase from Invitrogen [ThermoFisher Scientific, Waltham, MA, USA]. The amplification protocol was described by Clermont et al., 2000 [1], beginning with an initial denaturation at 94 °C for 5 min, followed by 30 cycles of 94 °C for 30 s (denaturation), 55 °C for 30 s (annealing), and 72 °C for 30 s (extension), as well as a final extension of 72 °C for 7 min. The PCR targets the genes *chuA* (279 bp), *yjaA* (211 bp), and *TspE4.C2* region (152 bp). This classification system identifies four primary phylogroups:Phylogroup A: Characterized by the absence of *chuA* and *yjaA* genes and typically does not amplify the *TspE4.C2* region.Phylogroup B1: Also lacks the *chuA* gene but may amplify the *TspE4.C2* region.Phylogroup B2: Positive for *chuA* and *yjaA* genes. May or may not amplify the *TspE4.C2* region.Phylogroup D: Positive for *chuA* but does not have the *yjaA* gene. May or may not amplify the *TspE4.C2* region.

Amplified products were analyzed on 3% agarose gels [Benchmark Scientific, Inc., 08872 Sayreville, NJ, USA] and visualized using the EpiChemi II Darkroom transilluminator UVP [Analytik Jena GmbH+Co. KG, 07745 Jena, Germany].

### 2.5. Detection of Virulence Genes

Eight *E. coli* virulence genes were analyzed using the multiplex PCR method described by Johnson and Stell, 2000 [22], using 100 ng of *E. coli* DNA, 0.6 μM of ach primer, and 0.6 U of Taq DNA polymerase [ThermoFisher Scientific]. The PCRs targeted the PAI region (the pathogenicity island described in virulent extra-intestinal *E. coli* strains), *pap*A (P fimbriae), *fim*H (type 1 fimbriae), *ibe*A (invasion of brain endothelium gene), *fyu*A (yersiniabactin siderophore receptor), *iut*A (aerobactin siderophore receptor), *hly*A (hemolysin A), and *tra*T (serum resistance gene), using the following PCR mixtures:PCR mix 1: PAI (925 bp), *pap*A (717 bp), *fim*H (508 bp), *ibe*A (171 bp);PCR mix 2: *fyu*A (787 bp), *iut*A (302 bp);PCR mix 3: *hly*A (1177 bp), *tra*T (290 bp).

The amplification protocol began with an initial denaturation at 94 °C for 5 min, followed by 30 cycles of 94 °C for 30 s (denaturation), 54 °C for 30 s (annealing), and 72 °C for 1.5 min (extension), with a final extension at 72 °C for 7 min. Amplified products were then separated on 3% agarose gels, stained with SYBR Green, and visualized under UV light using the EpiChemi II Darkroom transilluminator UVP.

### 2.6. Clonal Classification

All *E. coli* clinical isolates were analyzed and classified into clades by pulsed-field gel electrophoresis (PFGE) using a modified PulseNet protocol as follows [23]. *E. coli* isolates were cultured on blood agar base [Becton Dickinson and Company Corporate BD, 07417-1880 Franklin lakes, NJ, USA] and incubated at 37 °C for 24 h. Bacteria were then resuspended in PIV buffer (Tris-HCl 2 M pH 7.6, NaCl 5 M) [Merk KGaA] and washed three times. The bacterial suspension was adjusted to 0.5 McFarland Standard [ThermoFisher Scientific] in TE 1X buffer [ThermoFisher Scientific]. A volume of 150 µL of the adjusted suspension was embedded in 2% pulsed-field agarose (*v*/*v*) and solidified in plug molds [Bio-Rad Laboratories Inc., 94547 Hercules, CA, USA].

Agarose-embedded DNA lysis was achieved by immersing the plugs in 1 mL of lysis buffer (2 M Tris-HCl, pH 7.6; 5 M NaCl; 0.5 M EDTA [Merk KGaA], pH 7.6; 5% Brij58; 10% deoxycholate [ThermoFisher Scientific]; 20% sarcosyl [Merk KGaA]; 10 mg/mL lysozyme [ThermoFisher Scientific]) at 37 °C for 12 h with slow agitation. This was followed by treatment with ESP solution (0.5 M EDTA, pH 9; 20% sarcosyl; 20 mg/mL proteinase K [ThermoFisher Scientific]) at 50 °C for 12 h, again with slow agitation. The agarose plugs were then washed three times with ultrapure water [ThermoFisher Scientific] at 50 °C for 15 min each, followed by three washes in 1X TE buffer at 50 °C for 15 min each.

For DNA restriction, the agarose plugs were first incubated in a pre-digestion solution (1X XbaI buffer and 1% bovine serum albumin [BSA]) at 37 °C for 30 min. They were then transferred to a fresh restriction reaction mixture containing 1X XbaI buffer, 1% BSA, and 30 U of XbaI from Invitrogen [ThermoFisher Scientific, 02451 Waltham, MA, USA] and incubated at 37 °C for 3 h.

PFGE was performed in 1% pulsed-field agarose using the CHEF Mapper^®^ XA System [Bio-Rad Laboratories Inc.] with the following settings: 6 volts, 120° angle, initial pulse of 2.2 s and final pulse of 54.2 s, linear factor, and at 14 °C for 24 h. The restriction patterns were digitized in the EpiChemi II Darkroom [Analytik Jena], and DNA profiles were analyzed using Bionumerics^®^ version 7.6 software [Applied Maths, a Biomerieux Company, 9830 Sint-Martens-Latem, Belgium]. The analysis included cluster analysis using the unweighted pair group method with arithmetic mean (UPGMA) band based on the Dice coefficient. Optimization was set at 1%, band filtering was applied with a minimum height of 1% and a minimum surface of 1%, and band matching was performed with a tolerance of 2%. These conditions ensured accurate comparison of DNA banding patterns and the establishment of clonal relationships among isolates.

### 2.7. Comparison and Statistical Analysis

Comparative and statistical analysis was based on Tenover’s criteria for outbreak classification. According to these criteria, isolates with 100% similarity are considered indistinguishable, while those with 85–90% similarity were likely part of the outbreak. A similarity of 70–80% suggests a possible association with the outbreak, while similarity below 65% is classified as unrelated. Fragments with molecular sizes between 48.5 kb and 485 kb were analyzed.

Association analyses were conducted between phylogenetic classifications, virulence gene counts, and factors such as extended-spectrum β-lactamase (ESBL) production and multidrug resistance profiles.

Statistical analyses were performed using IBM SPSS Statistics 27 [IBM, 10504-1722 Armonc, NY, USA]. Percentages and measures of central tendency were calculated. For non-parametric variables, the chi-square test was used, with a significance level of *p* < 0.05. In cases involving fewer than 5 data points, bilateral Fisher’s exact test was applied, with a significance level of *p* < 0.05. A *p*-value < 0.05 indicated a significant association with our independent variables. Analyses with missing data were excluded. Risk was assessed using odds ratio (OR) analysis based on 2 × 2 contingency tables, with a 95% confidence interval (CI 95%).

Graphs were generated using the ForestPlotter package version 1.1.1 in the R platform.

## 3. Results

Between January 2017 and December 2019, 49 neonates diagnosed with sepsis were identified and treated at INPer. These neonates were preterm, with gestational ages ranging from 26 to 36 weeks. Some patients had two separate isolations of *E. coli* at different times, resulting in a total of 52 isolates. The distribution of isolates by year was as follows: 14 in 2017, 20 in 2018, and 18 in 2019. Most isolates were from blood and urine samples.

Once the identity of the *E. coli* strains was confirmed by PCR, drug susceptibility testing and gene amplification were performed. Phylogroup A was the most frequently identified, followed by phylogroup B2. Interestingly, the isolation frequency of phylogroup A strains increased over time. Regarding virulence genes, *fim*H and *fyu*A were the most commonly detected (Table 1). Notably, the presence of extensively drug-resistant (XDR) strains was little higher than strains sensitive to antibiotics (Table 1).

The isolates were categorized according to specific criteria: β-lactamase (ESBL) production, extremely drug-resistant (XDR) status, number of virulence genes (1–3 genes classified as “Low virulence strains” and 4–8 genes as “High virulence strains”), and the classical classification of the virulent phylogroups B2/D. The goal was to explore associations between neonatal or maternal clinical variables with the molecular characteristics of the isolates, as well as to analyze potential relationships among the molecular features of *E. coli*.

### 3.1. Association with ESBL Production

Analysis of the molecular characteristics of *E. coli* based on ESBL production revealed significant associations with several genes: PAI region (*p* = 0.019; OR 5.089; 95% CI 1.385–18.696), *pap*A (*p* = 0.006; OR 5.079; 95% CI 1.551–16.638), *iut*A (*p* = 0.033; OR 4.706; 95% CI 1.137–19.484), and *tra*T (*p* = 0.033; OR 4.706; 95% CI 1.137–19.484). These findings indicate that ESBL-producing strains have an 82% probability of carrying the *iut*A and *tra*T genes and an 83% probability of carrying the PAI and *pap*A regions, all of which are associated with virulence (Figure 1).

Additionally, ESBL production was significantly associated with strains belonging from phylogroups B2/D (*p* = 0.019; OR 4.014; 95% CI 1.223–13.173) (Figure 1). This suggests that B2/D strains have an 80% increased likelihood of producing ESBL.

Among clinical variables, non-ESBL-producing strains were associated with a 53% increased risk of chorioamnionitis. While clinically relevant, the *p* value was not significant and confidence intervals crossed 1, indicating the need for a larger sample size to confirm this association (Figure 1).

### 3.2. Association with XDR Profile

The analysis of XDR *E. coli* strains revealed a significant association with PAI (*p* = 0.005; OR 10.625; 95% CI 1.874–60.246), *pap*A (*p* < 0.001; OR 13.125; 95% CI 2.743–62.813), and *tra*T (*p* = 0.008; OR 14.400; 95% CI 1.567–132.311). The findings suggest these genes are more frequently present in XDR strains, with a 91–93% increased risk compared to antibiotic-sensitive strains (Figure 2). No associations were found with maternal or neonatal clinical variables.

### 3.3. Association with Virulence

For strains categorized by the number of virulence genes, significant associations were found in “High virulence strains” with PAI (*p* < 0.001; OR 69.33; 95% CI 7.583–633.904), *fim*H (*p* < 0.001; OR: 27.75; 95% CI 3.917–263.472), *fyu*A (*p* < 0.001; OR 37.00; 95% CI 3.917–349.515), and *iut*A (*p* < 0.001; OR 9.60; 95% CI 2.371–38.866) (Figure 3). These findings suggest an increased likelihood of these genes being present in strains classified as “High virulence”: *iut*A (90%), *fim*H (96%), *fyu*A (97%), and PAI (98%). In terms of maternal outcomes, “Low virulence strains” were associated with an 87% increased risk of developing cervicovaginitis and an 83% increased risk of maternal urinary tract infections.

### 3.4. Association with Phylogroups B2/D

When analyzing the molecular characteristics associated with phylogroups B2/D, significant associations were found with *iut*A (*p* = 0.038; OR: 3.692; 95% CI 1.039–13.122) and *tra*T (*p* = 0.013; OR 5.729; 95% CI 1.507–21.78). These results suggest a 78% increased probability of *iut*A presence and an 85% probability for *tra*T within phylogroups B2/D (Figure 4).

Conversely, in cases where phylogroups A/B1 were isolated, genes *hly*A (*p* = 0.044) and *ibe*A (*p* = 0.016) were detected in these traditionally non-virulent phylogroups, with increased probabilities of 82% and 92%, respectively. Neonatal outcomes associated with isolates from phylogroup A/B1 indicated an 82% increased risk of fever (*p* = 0.027), a 79% risk of tachycardia (*p* = 0.033), and an 82% risk of LBW or VLBW (*p* = 0.019).

For the epidemiological analysis of *E. coli* strains using pulsed-field gel electrophoresis (PFGE), Tenover’s criteria were applied, requiring at least 20 bands for analysis using the XbaI restriction enzyme. Similarity percentages were calculated based on the number of differing bands.

Of the 52 strains analyzed by PFGE, 6 distinct clusters were identified as “possibly related” to the same outbreak. Notably, two main branches of circulating hospital strains (designated as branches A and C) were distinguishable, both present between 2017 and 2019 (Figure 5). Each outbreak occurred independently but within the same time period, as evidenced by several shared isolation sites. In branch A, the majority of isolates belonged to phylogroups A and B2, with most of the strains (24/52) in this group, of which 18 exhibited significant multidrug resistance and were isolated from critical care neonatal units. In contrast, branch C predominantly consisted of strains from phylogroups A and D (12/52), with only 3/12 showing multidrug resistance patterns (Figure 5).

When analyzing the potential associations between the two major clusters and clinical or molecular variables, strains from branch A showed a significant correlation with a high virulence profile (*p* = 0.011) and were more likely to harbor the PAI and *iut*A virulence genes (Figure 6A). Specifically, strains from branch A were 89.2% more likely to possess between four and eight virulence genes, with probabilities of 79.2% and 90.5% for carrying the PAI and *iut*A genes, respectively. Additionally, maternal chorioamnionitis emerged as a significant risk factor for infections caused by branch C strains, with an 89% increased likelihood of contracting these (*p* = 0.016; OR 8.33) (Figure 6B).

## 4. Discussion

Neonatal sepsis remains a significant concern in preterm infants, with *E. coli* identified as one of the most common and relevant pathogens in this population at the INPer. Our study analyzed 49 preterm neonates, isolating 52 strains of *E. coli*. Conducted in a tertiary healthcare facility specializing in high-risk pregnancies and preterm deliveries, this research may not fully represent the general population pattern of neonatal sepsis. However, it offers valuable insights into the molecular biology of infection in high-risk neonates. The molecular pattern shown in our analyzed data makes it imperative to begin this discussion by focusing on the virulence factors associated with the strains.

Among the isolated strains, those typically classified as commensal *E. coli* (phylogroups A/B1) exhibited a concerning correlation with higher pathogenicity in our population. Notably, two virulent genes were predominantly observed: invasion of brain endothelium protein A (*ibe*A) and Alpha-hemolysin toxin (*hly*A). While *ibe*A is not the sole invasion protein required for complete *E. coli* penetration into the bloodstream and the blood–brain barrier, previous studies by Huang et al. and Xu et al. underscore the significance of both *ibe*A and its receptor (*ibe*A*R*) in facilitating rapid spread through cerebro spinal fluid, which heightens the risk of meningitis and subsequent neonatal sepsis in this population [24,25]. Although the number of meningitis cases in our study was not statistically significant, there was a notable trend towards higher rates of meningitis in the A/B1 group.

Furthermore, Barcellini et al. highlight that *E. coli* strains possessing the *hly*A gene are more likely to induce bacteremia, leading to neonatal sepsis [26]. This finding aligns with the work of Chambers et al., which emphasized the importance of P2X receptors in relation to *hly*A in *E. coli* strains associated with meningitis. The *hly*A protein facilitates cellular membrane breakdown, thereby promoting the spread of bacteria and resulting in systemic dysregulation, ultimately culminating in neonatal sepsis. Also, in patients expressing P2X receptors on macrophages, the *hly*A-mediated apoptosis of these immune cells may lead to reduced inflammation, enhanced control of infection, and improved survival rated for affected newborns [27].

Meanwhile, the phylogroups B2/D exhibited a stronger correlation with the carriage of the aerobactin gene, *iut*A and the complement resistance gene, *tra*T. In the study conducted by D’Onofrio et al., results showed a higher incidence of carriage of the *iut*A gene in *E. coli* isolates from adult patients with sepsis, with 58.8% of those cases associated with death as an adverse outcome. This demonstrates a pattern of higher virulence and severe disease in comparison with cases that did not carry the gene [28]. In another study conducted on Mozambican children with *E. coli* bacteremia, a direct correlation was found between the carriage of both *tra*T and *iut*A and the development of highly resistant and virulent strains, which showed an increased probability of severe clinical manifestations [29]. A study in children with septicemia found a higher incidence of *ibe*A, *iut*A, *hly*A, and *tra*T genes, specifically associated with the B2 phylogroup [30]. In a study conducted by Daga et al., the phylogroup with the highest number of virulence factors, associated with higher prevalence among the strains isolated from patients suffering from bacteremia, including some neonatal sepsis cases, was phylogroup B2 [31].

In our study, we observed a higher incidence of fever and tachycardia among the A/B1 phylogroups. It is well known that the phylogroups more pathogenic and symptomatic in humans are the B2 and D [1,2,3]. A study by Kobayashi et al. reported a higher prevalence of B2 as the main phylogroup causing infections in adults, with direct correlations to severe clinical findings [32]. To date, no data explain the correlation between the phylogroups and the development of clinical findings in neonatal sepsis. Patterns of adverse outcomes in neonatal sepsis vary by population. For example, Chinese neonates, the most significant complications included extremely low of birth weight, respiratory distress, and seizures. Italian neonates were more affected by very low birth weight, pneumonia, and mechanical ventilation [33,34]. In Swiss neonates, a higher incidence of neurodevelopment impairment was reported, while American neonates predominantly experience postnatal growth failure, increase inflammation and nutritional deficiencies [35].

As has been established, phylogroups B2 and D are globally more prevalent in causing infections across age groups, including newborns, children, and adults. Our population, however, did not follow this pattern, showing a higher prevalence of A and B1 phylogroups in septic newborns. This discrepancy underscores the need for further research to understand the impact of these phylogroups on neonatal sepsis in this population, as it may influence the way we study and treat this disease. Furthermore, the genes identified in these phylogroups are highly associated with dissemination and resistance, presenting a healthcare challenge that needs to be addressed to reduce morbidity and mortality.

Among the analyzed data to identify possible factors influencing this pattern, we found a correlation between the A and B1 phylogroups and newborns with LBW and VLBW. In a study conducted in Indian septic neonates, it was established that newborns with LBW, VLBW or extremely low birth weight (ELBW) may experience at least one infection episode. These infections may lead to bacteremia and subsequent neonatal sepsis. Their study showed that these conditions could impact the normal neurodevelopment of preterm neonates, affecting further growth and development [36].

Furthermore, a study by You et al. found a correlation between neonatal sepsis, respiratory distress syndrome, and LBW in neonates [37]. Although our population did not show a direct increase in tachypnea or respiratory distress syndrome, it is clear there is a correlation between the clinical severity of LBW and VLBW neonates, potentially linked to the reduced adaptability of these preterm neonates.

In our analysis, we categorized the virulence of *E. coli* strains based on the presence of eight virulence genes, employing a cut-off from four to eight virulence genes to identify the high virulent strains. Strains harboring one to three genes were classified as lower virulent strains. The more virulent strains contained *fim*H, which encodes the fimbriae protein that facilitates adherence, thus enhancing the survival and dissemination of *E. coli* in neonates, and *fuy*A, which encodes for ferric yersiniabactin uptake, aiding in metabolic processes and growth [4]. Khan et al. demonstrated a strong association between *fim*H and bloodstream dissemination of *E. coli* strains, particularly in brain microvascular epithelial cells. They concluded that *fim*H’s function extends beyond adhesion, enabling the spread that contributes to the dysregulation observed in neonatal sepsis [38]. Although the incidence of meningitis was not significantly higher in the most virulent group, it indicated a pattern of increased meningitis rates in premature neonates with sepsis compared to their less virulent counterparts.

Moreover, Burdet et al. reported that more virulent strains found in patients under three months with bacteremia were carriers of the gene *fyu*A, crucial for iron metabolism and survival in the bloodstream [39]. This finding reinforces our results, indicating that the most virulent strains harbored these genes, which foster a pro-inflammatory state that facilitates bacterial dissemination. The combination of these genes enhances the bacteria’s ability to adhere, spread, and survive, complicating eradication efforts and increasing mortality rates.

Furthermore, we found that “high virulence strains” were also carriers of the PAI region (*p* < 0.001; OR 69.330; 95% CI 7.583–633.904), a pathogenicity island, and *iut*A gene (*p* < 0.001; OR 9.6; 95% CI 2.371–38.866), which encodes for the ferric aerobactin receptor. Barcellini et al. also analyzed virulence factors among *E. coli* strains and found a greater number of virulence factors in strains associated with the onset of infection and sepsis during the first 24 h of life in neonates, particularly *iut*A [26]. Similarly, Guo et al. observed a significant pattern among infected neonates across six different hospitals in China, noting an increased number of virulence factors in neonates with invasive infection, especially from the genes *fim*H, *iut*A, and *ksp*MT II [40].

Although the results of our study did not show a statistically significant association between phylogroups B2 and D and a higher risk of increased virulence factors, a notable pattern emerged, with 76% of high virulence strains belonging to these phylogroups, compared with A and B1. Barcellini et al. reported similar findings, demonstrating an association between B2 and D phylogroups and the presence of multiple virulence factors (two to five genes), particularly *pap*A and *iut*A [27].

Our study also revealed a clinical correlation between maternal outcomes and “low virulent strains”, with associations found between low virulence strains and maternal conditions such as cervicovaginitis (*p* = 0.020; OR 6.708; 95% CI 1.464–30.733) and urinary tract infection (UTI) (*p* = 0.019; OR 5.056; 95% CI 1.239–20.626), before their delivery. Conversely, Forson et al. conducted a molecular analysis on strains from urinary samples of pregnant women with UTIs, showing that highly virulent strains were linked to pathogenic and resistant UTIs, particularly during the second and third trimester, which may pose a risk for neonatal infections [41]. In terms of cervicovaginits, Cook et al. analyzed *E. coli* strains isolated from the reproductive tracts of pregnant women with cervicovaginits whose newborns subsequently developed neonatal sepsis. They found an association between high virulence strains and these maternal infections, especially with genes like *MRHA*, *pap*, and *hly* [42]. Further research is needed to establish stronger correlations between these maternal outcomes and their role as risk factors for neonatal sepsis caused by low virulence strains.

Our molecular analysis also revealed a significant correlation between virulence factors and antibiotic resistance. Specifically, we observed a higher prevalence of *rpa*I, a pathogenicity island that encodes a plasminogen activator inhibitor, which aids in immune evasion and promotes colonization during infection, as well as *pap*A, responsible for pili subunit synthesis, facilitating adhesion and increasing pathogenicity. Additionally, *tra*T, which assists in immune evasion, and *iut*A, responsible for aerobactin synthesis, which aids in iron uptake, were prevalent in extended-spectrum β-lactamase (ESBL) strains, indicating a clear connection between virulence and resistance profiles. These virulence factors were also present in extremely drug-resistant (XDR) strains, excluding the *iut*A gene.

Fatima et al. highlighted the significance of these and other genes in contributing to increased resistance within a Pakistani population, particularly concerning urinary tract infections. Their findings illustrate how these encoded genes facilitate resistance mechanisms, promoting adherence and immune evasion while enabling expression of resistance genes [39]. Additionally, Martinez- Santos et al., emphasized the correlation between these virulence genes and heightened resistance, complicating bacterial eradication and resulting in more severe symptomatic cases [43]. In our cohort, most patients exhibited more severe symptoms, including a high risk of apnea, and were more symptomatic; while not statistically significant, these findings were clinically relevant. Finally, *tra*T’s role in allowing *E. coli* strains to resist complement-mediated lysis has been associated with high-virulence phylogroups such as B2 and D. Radera et al. demonstrated a direct correlation between multidrug resistance and the presence of *tra*T in similar *E. coli* patterns [44].

We also observed that ESBL-producing strains were significantly associated with the B2 and D phylogroups, highlighting the importance of identifying these phylogroups. Notably, 80% of strains belonging to phylogroups B2 and D were ESBL producers. Barcellini et al. found a similar association between ESBL strains with phylogroups B2 and D, particularly in highly virulent strains carrying the *iut*A gene [26]. Guo et al. also reported a correlation between the presence of multiple virulence factors and increased antibiotic resistance, specifically within high virulence strains of phylogroups B2 and D. Our findings, in alignment with these studies, further underscore the established link between higher virulence and resistance, contributing to the severe pathogenicity observed in infected and septic neonates [42].

PFGE analysis revealed significant genomic diversity among *E. coli* strains circulating within the Institute, identifying two major clusters, one of which was strongly associated with maternal chorioamnionitis. While this study provides insight into a localized perspective on epidemiology, it underscores the utility of PFGE in detecting potential outbreaks and uncovering associations with clinical risk factors and outcomes. These findings support the development of targeted prevention strategies tailored to the specific epidemiological context.

The application of PFGE in clinical studies has proven valuable in understanding strain dynamics. For instance, Jimenez-Rojas et al. (2024) [45] reported 35 distinct pulse types in a study of gut-colonizing ESBL-producing *E. coli* in neonates. One restriction pattern encompassed 37 strains isolated from 10 patients, half of whom developed healthcare-associated infections. This demonstrates the method’s effectiveness in detecting colonizing strains potentially implicated in infectious outcomes. Similarly, our findings suggest a higher probability of clonal dissemination, with the same strains persisting across different years, highlighting the importance of ongoing surveillance to mitigate clonal spread and associated risks.

The defense mechanisms available to neonates against infectious agents are limited. Due to their brief time ex utero, their immune system relies prominently on an immature innate immune response, which is fast but non-specific, allowing bacteria to spread more easily. The first and most critical factor facilitating bacterial infections is the loss of intestinal barrier integrity when infections are acquired gastrointestinally. Preterm neonates, in particular, are at increased risk of bacterial translocation. Studies have identified that increase in lactoferrin in the intestinal environment promotes the proliferation of *Lactobacillus* spp., which fortifies the intestinal barrier and prevents bacterial translocation [19].

Once a neonate becomes infected, the bacteria bypass the epithelial barriers, associated with molecular patterns (PAMPs) and are subsequently detected by pathogen recognition receptors (PRRs), such as the toll-like receptors (TLRs). This detection triggers the release of various cytokines, chemokines, complement proteins, and coagulation factors. Compared to adults, neonates produce significantly lower levels of pro-inflammatory cytokines, including TNF-α, IFN-γ, IL-1β, IL-6, and IL-8, which may be attributed to decreased production of intercellular mediators for TLR signaling. Additionally, neonates exhibit higher levels of anti-inflammatory cytokine IL-27, which has been associated with higher mortality, reduced weight gain, and poorer bacterial control in animal models. These outcomes stem from an imbalance between inflammation and bacterial survival, resulting in increased systemic inflammation [17,46].

Neutrophils, which play a pivotal role in initiating inflammatory responses in sepsis, differ markedly in neonates compared to adults, further weakening the immune response. Animal and human models have shown that neonates have a lower baseline neutrophil count, deceased expression of adhesion molecules (hindering migration), and reduced deformability. In septic conditions, these deficiencies, compounded by hypotension, can result in microvascular damage and potential organ dysfunction. Furthermore, neonates exhibit reduced neutrophil extracellular trap (NET) formation, diminished production of bactericidal proteins, and impaired apoptotic stimulation, leading to extensive tissue damage and further bacterial dissemination [19,46].

Compared to adults, neonates have a higher proportion of T cells, B cells, Natural Killer cells, and monocytes, but these cells differ in surface marker expression and antigen presenting capabilities. These differences contribute to difficulties in bacterial control and elimination. Additionally, neonates show reduced Th1 cytokine production, increased anti-inflammatory cytokines, and Th2 polarizing activity [17]. Collectively, these immune characteristics contribute to greater bacterial dissemination and an increased risk of sepsis, leaving neonates particularly vulnerable to invasive, highly virulent, and resistant pathogens.

## 5. Conclusions

In conclusion, our findings highlight the intricate relationship between virulence factors, antibiotic resistance, and *E. coli* phylogroups associated with neonatal sepsis in preterm newborns. Our findings reveal that phylogroups B2 and D are frequently associated with high virulence genes, aligning with traditional knowledge about these groups’ pathogenicity. Understanding these relationships is crucial for developing targeted interventions and treatment strategies to mitigate the impact of this serious conduction in such a vulnerable population. However, we also observed a surprising virulence potential in phylogroups A and B1, typically considered commensal strains. Their broad distribution of virulence genes and pathogenicity emphasizes the need to consider both phylogroups and virulence gene count (level of virulence) to accurately assess pathogenic potential, as focusing solely on phylogroups may overlook critical risks in vulnerable populations such as ours.

High virulence strains were associated with more severe outcomes, indicating the critical need for close monitoring and early intervention in infections involving these strains. The significant overlap between virulence factors and antibiotic resistance genes in ESBL and XDR strains further highlights the challenges in treating infections with limited antibiotic options and the importance of strict infection control practices.

Our findings challenge the established view of phylogroups A and B1 as being harmless, urging healthcare systems to broaden surveillance to include these groups, particularly in neonates with LBW and VLBW, who demonstrated heightened susceptibility in our study. By shifting the focus from traditional phylogroup classification to a more nuanced approach that incorporates virulence gene profiling, healthcare providers can enhance early detection, improve targeted interventions, and potentially reduce the mortality associated with neonatal sepsis. Further research should continue to refine these associations, offering new insights into bacterial pathogenicity and supporting the development of effective clinical management strategies of at risk neonatal populations.

## Figures and Tables

**Figure 1 microorganisms-13-00219-f001:**
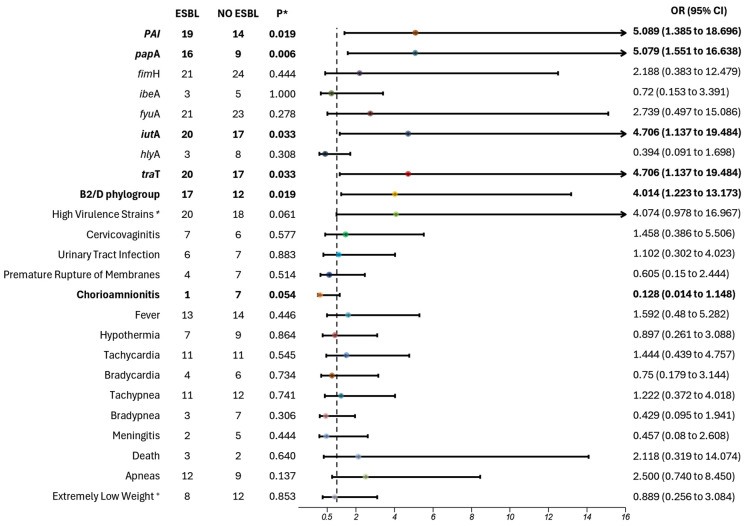
Extended-spectrum β-lactamase association with clinical and molecular variables of *Escherichia coli* in adolescent pregnant patients. ESBL production was assessed using the VITEK instrument, followed by an analysis to determine whether any variables were associated with ESBL-positive strains. Variables with potential associations are highlighted in bold. The dotted line represents the value of 1 on the x-axis. Horizontal lines that cross the value of 1 indicate no significant risk association. Extended-spectrum β-lactamase; NO ESBL—no extended-spectrum β-lactamase; OR—odds ratio; CI: confidence interval; * *p* < 0.05; ^+^ Extremely Low Weight vs. Very Low and Low Weight; ^≠^ strains with 4–8 virulence genes.

**Figure 2 microorganisms-13-00219-f002:**
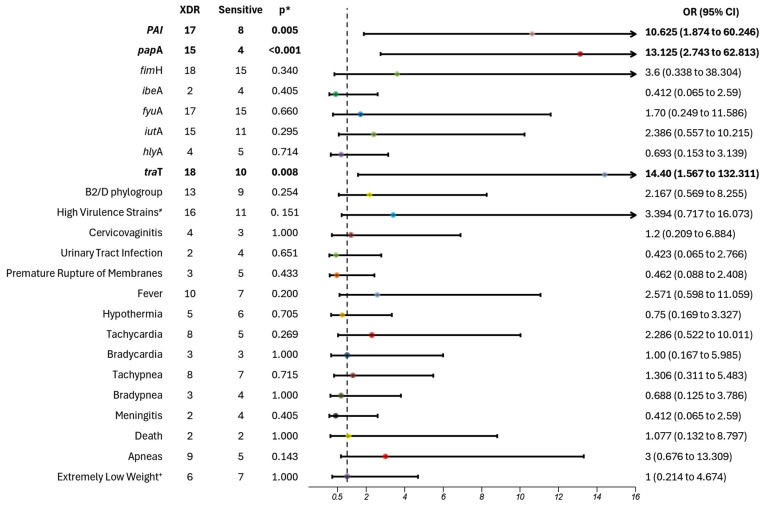
Extreme drug resistance association with clinical and molecular variables of *Escherichia coli* in adolescent pregnant patients. An association analysis was conducted to evaluate the relationship between various variables and the most resistant strains (XDR). The resistance profile was assessed using the WHO-Net free platform. Associations are highlighted in bold. The dotted line represents the value of 1 on the x-axis. Horizontal lines that cross the value of 1 indicate no significant risk association. XDR—extremely drug resistant; OR—odds ratio; CI: confidence interval; * *p* < 0.05; ^+^ Extremely Low Weight vs. Very Low and Low Weight; ^≠^ strains with 4–8 virulence genes.

**Figure 3 microorganisms-13-00219-f003:**
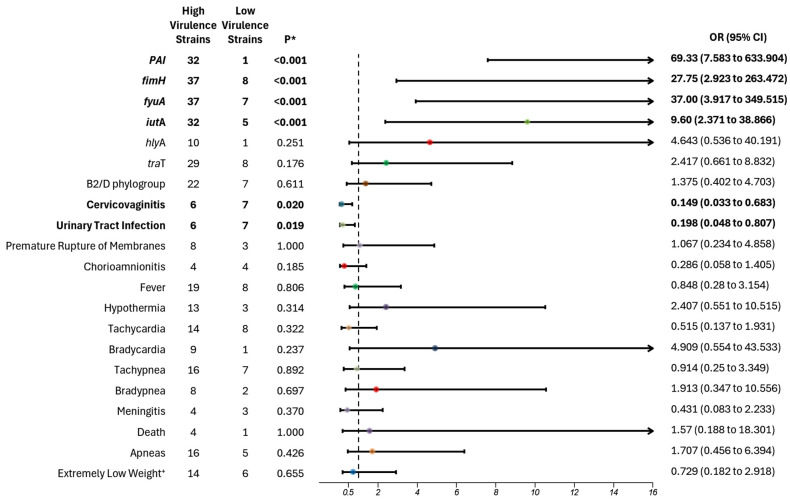
Virulence association with clinical and molecular variables of *Escherichia coli* in adolescent pregnant patients. High-virulence strains were defined as those carrying 4–8 virulence genes, while low-virulence strains carried 1–3 virulence genes. Four virulence genes were statistically associated with high-virulence strains (depicted on the right side of 1 on the *x*-axis in the graph), whereas two clinical variables were associated with low-virulence strains (below 1 on the *x*-axis). All variables with statistical significance are highlighted in bold. The dotted line represents the value of 1 on the x-axis. Horizontal lines that cross the value of 1 indicate no significant risk association. OR—odds ratio; CI: confidence interval; * *p* < 0.05; ^+^ Extremely Low Weight vs. Very Low and Low Weight.

**Figure 4 microorganisms-13-00219-f004:**
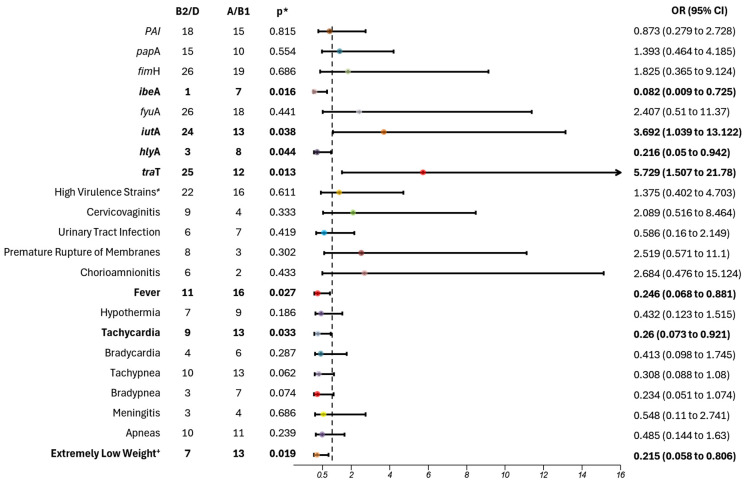
Phylogroup association with clinical and molecular variables of *Escherichia coli* in adolescent pregnant patients. Phylogroup classification was assessed based on the traditional distinction between virulent strains (B2 and D) and commensal strains (A and B1). Two virulence genes were statistically associated with B2/D strains (shown on the right side of 1 on the *x*-axis in the graph), while two virulence genes and three clinical variables were associated with A/B1 strains (below 1 on the *x*-axis). All variables with statistical significance are highlighted in bold. The dotted line represents the value of 1 on the x-axis. Horizontal lines that cross the value of 1 indicate no significant risk association. OR—odds ratio; CI: confidence interval; * *p* < 0.05; ^+^ Extremely Low Weight vs. Very Low and Low Weight; ^≠^ strains with 4–8 virulence genes.

**Figure 5 microorganisms-13-00219-f005:**
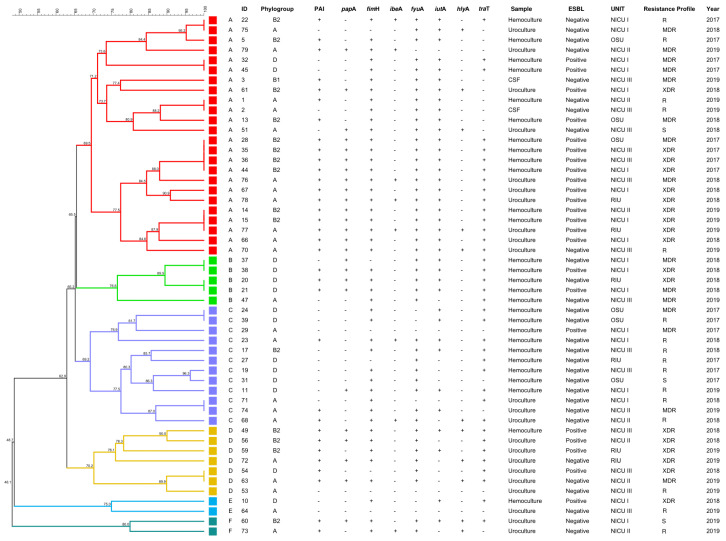
Clonality assessment of *Escherichia coli* strains with molecular variables in adolescent pregnant patients. Six clusters (A to F) were identified, with clusters A and C comprising 69% of the total strains. The figure illustrates the phylogroups, presence of virulence genes, resistance profiles, and, notably, the year of isolation, highlighting the prevalence of certain strains over time. ID—identification number; ESBL—extended-spectrum β-lactamase; NICU I—neonatal intensive care unit I; NICU II—neonatal intensive care unit II; NICU III—neonatal intensive care unit III; OSU—Obstetric-Surgery Unit; RIU—Rooming-In-Unit; R—resistant; S—sensitive; XDR—extensively drug resistant; MDR—multidrug resistance; CSF—cerebrospinal fluid.

**Figure 6 microorganisms-13-00219-f006:**
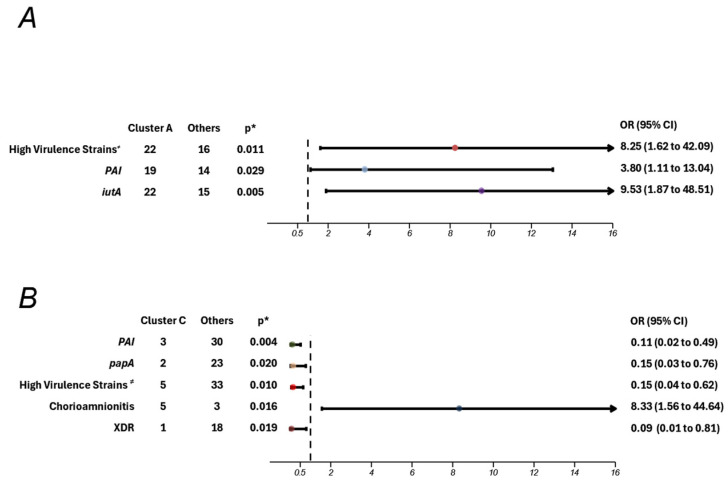
Cluster association to clinical and molecular variables of *Escherichia coli* in adolescent pregnant patients. (**A**) Cluster A association to molecular variables of *Escherichia coli* in adolescent pregnant patients. (**B**) Cluster C association to clinical and molecular variables of *Escherichia coli* in adolescent pregnant patients. The dotted line represents the value of 1 on the x-axis. OR—odds ratio; CI: confidence interval; * *p* < 0.05; ^≠^ strains with 4–8 virulence genes. This figure highlights only the variables with statistically significant differences, suggesting their association with strains from clusters A or C.

**Table 1 microorganisms-13-00219-t001:** Molecular characteristics and resistance profiles of *E. coli* strains.

		Year (*n*)			Total (*n*)
		2017	2018	2019	
Multi-Resistant Profile	XDR	3	10	6	19
	MDR	5	6	4	15
	Sensible	6	4	8	18
Phylogroup	B2	6	5	4	15
	D	7	6	1	14
	B1	0	0	1	1
	A	1	9	12	22
PAI		6	14	13	33
*pap*A		4	12	9	25
*fim*H		12	17	16	45
*ibe*A		1	4	3	8
*fyu*A		11	18	15	44
*iut*A		10	15	12	37
*hly*A		0	6	5	11
*tra*T		11	16	10	37

## Data Availability

Data are contained within the article, but for more details, please contact the corresponding author.

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
