# Peer review of "Advancing Understanding of Escherichia coli Pathogenicity in Preterm Neonatal Sepsis"

_microorganisms, 2025, doi:10.3390/microorganisms13020219_

Round 1
Reviewer 1 Report
Comments and Suggestions for Authors
The authors performed phylogenetic classification, virulence gene analysis, and antimicrobial resistance testing of the isolated strains. PFGEs are used to assess genetic correlations and outbreak clusters. Clinical data correlate with molecular findings. Phylogenetic groups A and B2 account for 46% of the strains. Phy-logroup A exhibits significant virulence with a high prevalence of rpai, iutA, and fyuA genes, along with significant drug resistance。There are some parts that need to be improved
1. Introduction, briefly explain why it is prone to infection with Escherichia coli leading to neonatal sepsis, and whether it comes from intestinal Escherichia coli.
2. Results: The words in Figure 1 are not very clear, make figure 1 clearer.
3. The same as figures 2-6, make figures 2-6 clear.
4. Discussion, Try to explore the mechanism by which Escherichia coli and its virulence factors cause neonatal sepsis, such as its relationship with innate immunity (ref:PMID: 29209311, PMID: 38615022).
5. Abstract, rpai, iutA, and fyuA, These genes need to provide their full names for the first time.
Materials and Methods, 2.5 “Detection of Virulence Genes”, Show some visualized results

Author Response
We thank the reviewer for their thoughtful revision of the manuscript. We believe that their comments have significantly enhanced the quality and clarity of our work.
Responses for the comments:
- Introduction, briefly explain why it is prone to infection with Escherichia coli leading to neonatal sepsis, and whether it comes from intestinal Escherichia coli.
AUTHORS RESPONSE: We included three detailed paragraphs on the subject in lines 97–118 to provide a comprehensive overview.
- Results: The words in Figure 1 are not very clear, make figure 1 clearer.
AUTHORS RESPONSE: The figure resolution has been enhanced, and a descriptive paragraph has been added to provide greater clarity.
- The same as figures 2-6, make figures 2-6 clear.
AUTHORS RESPONSE: Figures resolution has been enhanced, and a descriptive paragraph has been added to each one in order to provide greater clarity.
- Discussion, Try to explore the mechanism by which Escherichia coli and its virulence factors cause neonatal sepsis, such as its relationship with innate immunity (ref:PMID: 29209311, PMID: 38615022). Inmunidda recien nacido y premature ORLY
AUTHORS RESPONSE: We introduce the immunology of newborns and their susceptibility to infections in the introduction (lines 113–118) and provide a more detailed and clearer explanation in the discussion section (lines 593–629).
- Abstract, rpai- PAI, iutA, and fyuA, These genes need to provide their full names for the first time.
AUTHORS RESPONSE: We made the clarification of gene names (ln 33-35). Additionally, each gene product was mentioned in the Methods section (lines 194–199) for better understanding.
- Materials and Methods, 2.5 “Detection of Virulence Genes”, Show some visualized results
AUTHORS RESPONSE: The reviewer is correct; we did not previously present any separate results from the molecular characterization. We have now included a table (Table 1) summarizing the results of virulence gene identification, phylogroup classification, and the identification of XDR/MDR strains. These details have been added to the Results section in lines 269-274 and table inserted in line 277.
Reviewer 2 Report
Comments and Suggestions for Authors
microorganisms-3412583-peer-review-v1
The current study is interesting and reveal the role of different groups of E. coli in neonatal sepsis medical complications. Authors have collected significant numbers of E. coli and proceeded with their analysis. In my opinion paper deserve attention of the Editor and can be suggested for publication, however, some corrections, adjustments, update needs to be made by the authors.
Ln45: Gram needs to be with capital G, since is referring to the name of Hans Gram.
Some of the references are not really appropriately used in the text. Please, be sure when citing, you have link to appropriate reference. Example, on Ln 49 you have stated that Clermont et al. provided molecular classification, however, then providing 3 references, that will need to be cited different.
Ln55: E. coli needs to be in italics.
Please, when you introduce a species name for the first time, write it in full, and following occasions needs to be abbreviated.
Ln56: correct to: ...pathogenicity [1,4].
Ln59: Replace word beta with appropriate Greek symbol.
Ln76: Please, check the text: "4 to 15.4 per 1.00 live births". This text do not make sence. Do you want to say 1000?
Ln84: Maybe replace with streptococci or Streptococcus spp.
Ln90-91: Please, pay attention to Gram (with capital G) and italics for the microbial names.
For the suppliers of material and equipment, please, use the headquarters address and not local distributors. Example BioMerieux is a French company, not from USA. Moreover, for the suppliers, please, provide the name of the company, and address, including name of the city, state (in case of federal country) in abbreviated form and name of the country. IN following occasions, when same company was supplied different material or equipment, only name of the company is sufficient.
Please, provide a short description of the procedure applied in estimating MIC for the mentioned antibiotics. Antibiotics were provided from what supplier?
Maybe PCR conditions suggested from Clermont et al. (2000) can be described with some details in section 2.4. Same for section 2.5., please provide some details.
Better quality and resolution of picture 1 needs to be provided.
Results are presented well, and authors have tried to be as possible informative. However, the quality of the figures needs to be improved. Maybe different software can be used to build the figures?
Please, try not to repeat the results in the discussion section. And maybe look for some additional studies where E. coli was related to sepsis medical complications. Will be interesting to enrich the discussion with some additional examples from Latin American, and countries from around the Globe. The comparisons can be with considered developed countries, but as well with countries with similar incomes to that of Mexico.
References need to be formatted according to the recommendations form the Journal.
Author Response
We sincerely thank the reviewer for their insightful and constructive feedback. We believe that their comments have greatly improved the overall quality and depth of the manuscript.
Comment responses:
- Ln45: Gram needs to be with capital G, since is referring to the name of Hans Gram.
AUTHORS RESPONSE: Thank you very much for this observation, now is corrected in every place where Gram was mentioned.
- Some of the references are not really appropriately used in the text. Please, be sure when citing, you have link to appropriate reference. Example, on Ln 49 you have stated that Clermont et al. provided molecular classification, however, then providing 3 references, that will need to be cited different.
AUTHORS RESPONSE: We reviewed the document and identified additional citation errors, all of which have been corrected.
- Ln55: E. coli needs to be in italics.
AUTHORS RESPONSE: We are very sorry for this mistake. It was corrected.
- Please, when you introduce a species name for the first time, write it in full, and following occasions needs to be abbreviated.
AUTHORS RESPONSE: We have thoroughly reviewed the entire document to ensure all corrections were made.
- Ln56: correct to: ...pathogenicity [1,4].
AUTHORS RESPONSE: Thank you. It was corrected.
- Ln59: Replace word beta with appropriate Greek symbol.
AUTHORS RESPONSE: Greek symbol for beta was added to every place where it was mentioned.
- Ln76: Please, check the text: "4 to 15.4 per 1.00 live births". This text do not make sence. Do you want to say 1000?
AUTHORS RESPONSE: Thank you very much, a cero was missing, we wanted to say 1,000. Now is corrected (line 79).
- Ln84: Maybe replace with streptococci or Streptococcus spp.
AUTHORS RESPONSE: We decided to use streptococci. Now it is mentioned in line 87.
- Ln90-91: Please, pay attention to Gram (with capital G) and italics for the microbial names.
AUTHORS RESPONSE: We thank reviewer for this observation. Corrections were performed
- For the suppliers of material and equipment, please, use the headquarters address and not local distributors. Example BioMerieux is a French company, not from USA. Moreover, for the suppliers, please, provide the name of the company, and address, including name of the city, state (in case of federal country) in abbreviated form and name of the country. IN following occasions, when same company was supplied different material or equipment, only name of the company is sufficient.
AUTHORS RESPONSE: The reviewer is correct. We have now included the correct addresses of the headquarters of the material and equipment suppliers. These corrections have been highlighted in the Methods section.
- Please, provide a short description of the procedure applied in estimating MIC for the mentioned antibiotics. Antibiotics were provided from what supplier?
AUTHORS RESPONSE: We did not estimate the MIC for the antibiotics tested. We clarified this in lines 169-172 as follows: “Automated diagnostic panels (BioMérieux SA) were used for both blood and urine cultures, with automated antibiotic susceptibility testing specifically targeting Gram-negative bacteria. Resistance classification was performed following the 2023 CLSI guidelines using the WHONET platform.”
The BioMérieux panels include antibiotic concentrations established by the CLSI for resistance identification.
- Maybe PCR conditions suggested from Clermont et al. (2000) can be described with some details in section 2.4. Same for section 2.5., please provide some details.
AUTHORS RESPONSE: We did not include PCR conditions as they both (phylogroups and virulence genes identification) are techniques frequently reported, but now they are included in lines 174-180 for phylogroups: “E. coli isolates were classified into phylogenetic groups using 100 ng of DNA in the multiplex-PCR containing 0.125 M of each primer and 0.6U of Taq DNA Polymerase from Invitrogen (ThermoFisher Scientific). The amplification protocol was described by Clermont et al., 2000 [2], beginning with an initial denaturation at 94°C for 5 minutes, followed by 30 cycles of 94°C for 30 seconds (denaturation), 55°C for 30 seconds (anneal-ing), and 72°C for 30 seconds (extension), a final extension of 72°C for 7 minutes were performed.” For virulence genes amplification, PCR protocol was already mentioned but we included reaction mixture in lines 194-199.
- Better quality and resolution of picture 1 needs to be provided.
AUTHORS RESPONSE: Resolution of figures was improved.
- Results are presented well, and authors have tried to be as possible informative. However, the quality of the figures needs to be improved. Maybe different software can be used to build the figures?
AUTHORS RESPONSE: In deed, figures quality is very bad. We change the resolution of all of them, also we included a description of each one.
- Please, try not to repeat the results in the discussion section. And maybe look for some additional studies where coli was related to sepsis medical complications. Will be interesting to enrich the discussion with some additional examples from Latin American, and countries from around the Globe. The comparisons can be with considered developed countries, but as well with countries with similar incomes to that of Mexico.
AUTHORS RESPONSE: We removed the presentation of specific results from the discussion section to enhance readability and improve the flow of the narrative. Also, we added a paragraph in the discussion section (lines 468–474) addressing neonatal outcomes in various regions, including the Americas.
- References need to be formatted according to the recommendations form the Journal.
AUTHORS RESPONSE: The references have been thoroughly revised and improved. We utilized EndNote software, as recommended in the instructions for authors, and formatted the references using square brackets, adhering to the provided example. Additionally, we ensured that no references were placed after punctuation marks.
Round 2
Reviewer 1 Report
Comments and Suggestions for Authors
Within the scope of this investigation, the authors have provided sufficient detail and addressed the reviewer's comments